# Levels of Nature and Stress Response

**DOI:** 10.3390/bs8050049

**Published:** 2018-05-17

**Authors:** Alan Ewert, Yun Chang

**Affiliations:** 1School of Public Health, Indiana University, Bloomington, IN 47405, USA; 2School of Kinesiology and Recreation, Illinois State University, Normal, IL 61790, USA; ychan12@ilstu.edu

**Keywords:** biomarkers, human health, natural environments, psychological stress

## Abstract

A growing number of studies have shown that visiting green spaces and being exposed to natural environments can reduce psychological stress. A number of questions concerning the effects of natural environments on levels of stress remain including, “Are activities engaged in natural environments more or less beneficial at reducing stress when compared to those done in more urban settings?” This study examined this question from the perspective of “levels of nature”. That is, data on levels of stress were collected from three sites, one site having wilderness-like characteristics, a second site representing a municipal-type park, and a third site representing a built environment (indoor exercise facility) within a city. Data were generated using biophysical markers (cortisol and amylase) and a psychological measure within a pre- and post-visit format. Findings suggest that visiting natural environments can be beneficial in reducing both physical and psychological stress levels, with visitors to a natural environment reporting significantly lower levels of stress than their counterparts visiting a more urbanized outdoor setting or indoor exercise facility.

## 1. Background

A growing number of studies have shown that visiting green spaces and being exposed to natural environments can reduce psychological stress [1,2,3,4,5,6,7]. This fact alone is of increasing importance due to the many physical illnesses such as coronary disease and obesity that have been linked to issues often related to chronic stress [2,8,9]. For the purpose of this current study, stress is defined as a process through which specific demands (e.g., work, childcare, class assignments, etc.) are perceived as exceeding an individual’s resources or abilities to control or manage effectively. Moreover, it should be noted that the preponderance of research done in the area of human health and natural settings has generally involved people visiting a natural landscape and engaging in some type of physical or contemplative-based recreational activity (e.g., walking, sightseeing, fishing, etc.). This study expands on this body of knowledge by identifying what effect “level of nature” has on both self-reported and biometrically determined levels of stress by comparing changes in levels of stress from visitors to three different sites that varied on how closely their attributes represented a natural environment. These three sites included a wilderness-type setting, a municipal park, and a local fitness and recreation facility. 

Natural environments have been linked to the Attention Restoration Theory (ART) that proposes that these settings possess a particular set of properties that promote restoration from attention fatigue [10,11,12]. Similar to ART, the psycho-evolutionary theory (PET), proposed by Ulrich [13], posits that natural environments are effective at reducing levels of stress because they offer specific attributes that our species viewed as having inherent survival qualities, such as water and spatial openness. Hartig [14] integrates these two theories by suggesting that there is an “intertwining of the mechanisms” whereby the extent to which people are attracted to and use a natural environment is dependent on how restorative that specific environment is to them. Finally, and specifically related to stress-reduction outcomes, Degenhardt, Frick, Buchecker, and Gutscher [15] identified a number of variables such as state of health, self-efficacy, and quality of the neighborhood, that have a direct bearing on the frequency and type of use of natural environments.

In addition, within a phylogenic perspective (i.e., the evolutionary development), the underlying assumption of this study is that since human beings developed in natural environments these types of settings will be more “therapeutic” than those associated with built environments [7,16]. For example, Hartig and his colleagues [17,18] found that walking in a natural environment was more restorative than walking in urban surroundings. Similarly, Harte and Eifert [19] found running in the outdoors to be more effective at reducing negative emotions than running on a treadmill. Lee, Hur, Yang, Lee, and Lee [20] reported that visitation to forest environments could be beneficial to individuals suffering from a variety of ailments such as metabolic syndrome, while Gidlow et al. [21] report similar findings with walking in natural environments being linked to greater levels of restoration than urban settings. Stress represents the dichotomy between individual resources and specific demands that can result in the development of a number of undesirable physiological, psychological, behavioral, or social outcomes [22,23]. Recently, research has pointed to the effectiveness of reducing stress through physical exercise [24,25,26] and exposure to natural environments [7,13,17,18,27,28]. For example, Barton and Pretty [29] found physical activity performed in natural settings resulted in significant improvements in the mental health variables of self-esteem and mood. Likewise, Bowler, Buyung-Ali, Knight, and Pullin [30] found evidence of the direct and positive impacts on well-being and health from exposure to natural settings. However, there is still much to learn about how, and to what extent, these effects occur. One question salient to this discussion is what is the effect of the “type” of environment on levels of stress. More specifically, are activities engaged in natural environments more or less beneficial at reducing stress than those done in more urban settings? The purpose of this study was to measure the effect of visitation to one of three areas consisting of differing levels of nature (natural, semi-natural, urban) upon levels of stress using both physiological and psychological data collection.

There are a number of ways in which natural environments may promote human health by reducing stress. Natural environments can often provide the setting for physical activity, with numerous studies reporting the beneficial effects of “green” exercise [6]. Exercise in outdoor settings has been reported to be more restorative and stress-reductive than indoor exercise [31]. Walking in greenspaces and other outdoor settings has been linked to increases in self-esteem and overall mood levels [32]. Moreover, reported intentions to continue participation in walking exercises was higher in respondents using the outdoors when compared to those in indoor settings [33].

More specific to this study, stress responses have been studied from the perspective of location with the natural/urban dichotomy being the most subscribed to, and often within a laboratory type setting. For example, Ulrich and his colleagues [7] used a stressful movie followed by videotapes of natural and urban settings to measure stress. Their resultant data, based on physiological measures such as skin conductance, muscle tension, and pulse transit time, pointed to a faster and more complete recovery time from the stressful effects of the movie when participants were exposed to the natural landscapes scenes. To compare stress recovery in natural and urban field settings, Hartig and his colleagues [17] compared psychophysiological stress recovery using repeated measures of ambulatory blood pressure, emotion, and levels of attention on a split group of young adults. Their data consistently suggested more positive effects on stress reduction from the natural settings as opposed to the urban one. While not specifically focusing on stress reduction, after reviewing eleven different studies, Coon et al. [34] found that engaging in physical activity outdoors was more effective for enhancing feelings of revitalization, decreases in tension, and moderated levels of depression. Pasanen, Tyrvää and Korpela [35] lent support for these findings by providing data that natural outdoor settings were more effective than built indoor environments in enhancing emotional wellbeing. This finding was also suggested by Bowler, Buyung-Ali, Knight, and Pullin’s [30] review of 25 studies which also supported the contention that natural landscapes can be more effective than urban locations on a number of dependent variables such as emotions, level of attention, and physiological parameters including immune function and endocrine changes that reflected levels of stress. They also posited that comparing the “quality” of different natural environments could be important for future work, and this has implications for this study.

Of immediate importance to this study, is the work by Beil and Hanes [36] who examined the effect of visitation to different types of environments ranging from “very natural”, “mostly natural”, “mostly built”, to “very built” on levels of cortisol and α-amylase. Their findings supported the contention that natural settings were more effective than built settings in reducing levels of stress as measured by both cortisol and α-amylase. Although not specific to stress reduction, Lee et al. [20] found that visiting different forest types (wild forest versus a tended forest) for patients suffering from metabolic syndrome (MetS) produced marked differences in acute insulin response, pulse rate, and oxidative stress markers with the wild forest being associated with more positive health outcomes. This study sought to add to the body of knowledge by investigating the effects of visitation to different types of field-based environments, with varying degrees of nature, upon physiological and psychological-based measures on levels of stress.

## 2. Methods

### 2.1. Research Questions

Two biophysical markers and a psychological measure were used in this study to address the following research questions, after controlling for initial levels of stress, time of day, and weather. Using the previous literature, we made the following hypotheses which served to guide this study:

**Hypothesis** **1** **(H_1_).**
*When comparing pre- and post-visit physiological measures of cortisol and α-amylase, Site A, featuring the highest level of nature, will show the greatest levels of stress reduction.*


**Hypothesis** **2** **(H_2_).**
*When comparing pre- and post-visit psychological measurements using the PSQ, Site A (highest level of nature) will show the highest reduction in stress levels.*


### 2.2. Location

Within the framework of this work, three sites were utilized in this study, each with a different level of nature. Each site was in relatively close proximity to a medium-size city (est. population of 46,000 people) in the Midwest United States. The three locations consisted of: (Site A) a “natural” setting, (Site B) a municipal park, and (Site C) an urban “built” exercise facility.

Site A (natural setting) served as the setting with the highest level of nature and is a 1200-acre, wilderness-like forested area primarily comprised of trails that wind through wooded ridges and ravines. In this site, hiking and wildlife watching are important recreational endeavors, either individually or in small groups (see Figure 1a). It should be noted that the hiking trails utilized by the visitors to this site are located away (at least 50 m) and out of sight from the 100-acre lake present in the photograph. It was felt that this distancing from the water was important since neither Site B or C had a water resource and water can serve as an attractor and confounding variable in this case [37]. Site B (semi-natural) is a 33-acre municipal park featuring walking paths, places for gatherings, playgrounds, and open field space for causal recreational activities (see Figure 1b). Site C, (urban built) represented a full service indoor exercise facility featuring an indoor running track, tread masters, and free-weight areas (see Figure 1c)*.*

### 2.3. Sample Design

This study utilized a quasi-experimental pretest-posttest design. A questionnaire was administered to collect demographic data from randomly selected park users from all three sites in order to develop a baseline understanding of the population characteristics, such as age, sex, and type and level of physical activity engaged in by participants to each site. Adopting the purposive sampling method, participants displaying attributes of the aforementioned general characteristics found in the initial questionnaire were approached by study enumerators as they entered the setting, informed of the study, and asked for their participation. Upon agreeing to participate, participants were provided an informed consent form to participate through the IRB process determined by Indiana University (protocol number 1409227613). Following this process, participants were asked to provide physiological and psychological measures of stress levels. These data were collected by the researchers just before entering the site, and immediately upon ending their visitation to the site. To equate for activity type, for both Sites A and B, visitors were predominately hiking, while for Site C, the primary activity was running or walking on the indoor track.

### 2.4. Physiological/Biomarker Measures

While a broad spectrum of studies have reported positive physical and psychological effects relative to time spent in natural environments, most of these studies have been based on self-report measures, or in some cases, measures of parental report [30]. Physical measures have been less prevalent, but have included measures of endocrine health, cardiovascular health, and immune system functioning with the use of physiological biomarkers becoming increasingly popular in research done on variables such as stress [38]. For example, measures of stress hormones such as α-amylase and cortisol have previously been measured using blood samples, but the process of drawing blood has been shown to actually increase the level of stress hormones in the blood stream [39]. Additionally, measures such as blood tests can be intrusive and difficult to perform in, or near, natural settings, making it challenging to test responses to the natural environment during the actual period of exposure. Noto et al. suggest that salivary measures of stress hormones are not only more reliable than blood tests, but also that the hormone α-amylase correlates significantly with state-trait anxiety measures [39]. Like cortisol, Granger, Kivlighan, El-Sheikh, Gordis and Stroud [40] also support the view that α-amylase can be efficacious in identifying levels of psychobiological stress. As a result of these issues, the salivary measures used in this study were cortisol and α-amylase.

In stress-related research, however, the level of noninvasiveness and ease of sampling is of major importance for physiological biomarker measurements [41]. For this reason, saliva sampling or urine collection are often the measurement vectors of choice as opposed to serum or blood measurements. In this study, saliva samples were collected to test for changes in levels of the stress hormones cortisol and α-amylase using saliva collection vials. Study participants were asked to provide 3–5 mL saliva samples just prior to the start of their recreational experience and immediately following the conclusion of the experience using a drool method [42]. These samples were marked and stored frozen for later evaluation.

The first biomarker used in this study was cortisol, a steroid hormone, and belonging to a broader class of steroids called glucocorticoids which are produced by the adrenal gland and secreted during a stress response. A primary purpose of cortisol is to redistribute energy (glucose) to high priority parts of the body such as the heart, brain, and muscles and is often associated with the concept of fight, flight, or freeze [43]. Detrimental changes occur in the body if heightened levels of cortisol are present for extended periods of time, including the suppression of the immune system and muscle wastage. The negative effects are often associated with the term “chronic stress” and are associated with the body’s response to extended exposure to elevated levels of cortisol [44].

Cortisol was measured by ELISA techniques using a TECAN multi-plate reader. The sampling time frame for data collection was in accordance with that posited by Barker, Knisely, McCain and Best [45], which suggested that the optimal time for measuring salivary cortisol levels was within a 45-min time period following the activity. Because salivary cortisol levels are particularly subject to variability throughout the day, and particularly in the early morning awakening hours, as well as time of the last food consumption, only those respondents arriving to the study sites after 3:00 p.m. and had not eaten within two hours of their site visit were tested [46]. The experimental period for all three sites lasted from the middle of April through the middle of May. Individual saliva samples were collected into small test tubes, placed on dry ice, and then transported to a laboratory to be frozen at −80 °C until analyzed.

The second biomarker, α-amylase, is a protein enzyme that breaks down large polysaccharides such as starch and yields high energy glucose and maltose. Amylase is found in saliva and is secreted through activation of the sympathetic nervous system (SNS), often in response to events related to stress and perceived threat [47,48]. That is, by rapidly breaking down starches to sugars, easily accessible energy can be redistributed to parts of the body involved in the flight or fight syndrome such as the muscles, heart, and brain. Amylase is typically measured through samples of saliva and is thought to be a useful indicator of activity within the sympathetic nervous system (e.g., heart rate, pupil dilation, increased perspiration, etc.). In addition, amylase reacts and recovers more quickly from the stress event than cortisol, usually returning to baseline within 10 min post-stressor and can be affected by exercise and physical stress [49]. The α-amylase was measured by colorimetric approaches using a multiple spectrophotometer.

### 2.5. Perceived Stress Questionnaire (PSQ)

The Perceived Stress Questionnaire (PSQ) [50] measures four factors related to stress: worries, tension, joy, and demands, with five items for each factor. Internal consistently has a range of 0.90–0.92, and a test-retest value of 0.82 [51]. Cronbach alpha for the overall score is 0.85 and an overall reliability value of 0.80 [51]. Unlike other stress inventories, which often focus on external life events such as divorce or the death of a family member, the PSQ focuses primarily on internal stress reactions such as feelings of anxiety, exhaustion, frustration, or conflict, with only one scale, “demands”, focused on external stressors, such as time demands or deadlines. Subjects are asked to identify the intensity of their feelings on a 4-point Likert scale with 1 being “I almost never feel like this”, and 4 being “I usually feel like this”. This instrument has been validated using a variety of different samples, and measures both positive and negative aspects of stress.

### 2.6. Data Analysis

Data collected via demographic questionnaire were initially analyzed using descriptive statistics. In order to estimate the proportions of the users of the three locations, participants’ gender, average age, average visit time (measured by minutes) and frequency of visit per week (measured by one to three time vs. more than three times) were documented. Although there was some variance in the data, overall the demographic data were similar across all three sites. It was challenging to estimate an adequate sample size for this study because the covariance structure was unknown prior to obtaining the data. However, Fliege et al. [50] suggest at least 30 samples per cluster as an appropriate sample size and having a statistical power approaching 0.6 with a medium effect size of 0.5. Since there were three study sites, a minimum number of 90 participants was the target sample size. With no generally accepted guidelines concerning the empirically-derived meanings of the data currently available for the analyses of the biomarkers, the study should be considered exploratory given the low sample size.

## 3. Results

### 3.1. Descriptive Statistics

A total of 105 subjects was recruited from all three sites. Subjects include 63 males and 42 females. As expected, given the research design of purposively selecting participants, the demographic data are somewhat similar. As can be seen from Table 1, visitors to Site B had the highest average age (*M* = 37.2 years), followed by Site C (*M* = 28.8 years), and Site A (*M* = 25.9 years). Visitors stayed longest at Site B (*M* = 68.3 min), followed by Site C (*M* = 66.7 min), and Site A (*M* = 54.4 min). Most participants at Site A and Site C visited this recreational site one to three times per week. Visitors from Site B have a more bimodal attendance record between one to three times per week and more than three times per week. 

Visitors’ stress levels were measured using physiological stress and psychological stress indicators. In this study, cortisol and α-amylase levels were used to detect participants’ physiological stress levels and scores collected from PSQ scales were used to examine participants’ psychological stress levels.

### 3.2. Cortisol Levels

Using an ANOVA test to examine differences between visitors’ pre-test measurements of levels of cortisol from the three sites, the results show no significant differences among the three pre-test measurements (F_(2,101)_ = 0.67, *p* = 0.51). Paired sample *t*-tests were then conducted to compare participants’ stress levels before and after visiting the three different locations, respectively. As displayed in Table 2, the results indicated that participant’s cortisol levels significantly decreased after visiting Site A (natural) but not after visiting Sites B or C. When aggregated across all three sites, the results of the ANOVA test indicated that different locations did not have an overall significant impact on participants’ changes in cortisol levels (F_(2,95)_ = 1.86, *p* = 0.16). The effect sizes were considered low (0.01–0.04).

### 3.3. Amylase Levels

The result of the ANOVA test showed no differences between visitors’ pre-test measurements of levels of amylase from the three sites (F_(2,101)_ = 2.33, *p* = 0.10). Differences in pre- and post-visit levels of α-amylase showed inconsistent change of directions. As illustrated in Table 3, the results suggested that visitors experienced significant increases in levels of α-amylase after visiting Site C, but not after visiting Sites A or Site B.

This finding was supported by the results of the ANOVA test, which indicated that different locations had a significant impact on visitors’ changes in α-amylase levels (F_(2,101)_ = 3.36, *p* < 0.05). The post-hoc analysis using the Scheffe Method, revealed that visitors’ α-amylase levels were significantly higher after visiting Site C when compared with visiting Site B. However, no significant differences were between visitors from Site A and Site B, or between visitors from Sites A and Site C. The effect sizes for these comparisons were low (0.03–0.04).

### 3.4. Psychological Stress Levels

As depicted in Table 4 and Table 5, there were significant decreases in visitors’ levels of *demands* and *worries* after visitation to the respective three sites (*p* < 0.01). For levels of *tension*, none of the decreases measured at the three locations reached statistical significance (Table 6). For levels of *joy*, significant increases were observed at Sites A and Site B, but not from visitors to Site C (Table 7).

As shown in Table 7, the ANOVA tests indicated that there were differences found in visitors’ changes in levels of *joy* after visiting the three sites (*p* < 0.01). The post-hoc analysis, Scheffe’s Method, revealed that visitors have significant increases in levels of *joy* after visiting Site A, compared to visitors visiting Sites B and Site C. There were no differences found in visitors’ changes in levels of *demands* (*p* = 0.84), levels of *worries* (*p* = 0.06), and levels of *tension* (*p* = 0.27).

In summary, both H_1_ and H_2_ were partially supported. Among the three sites. each featuring a different level of nature, visitors to Site A (most natural) reported higher levels of stress reduction as measured by the variables of decreased levels of cortisol, demands, worries, and an increased level of joy, whereas Site B (semi-nature) visitors reported three indicators of reduced stress (decreased level of *demands* and *worries*, increased level of joys) and Site C (built) visitors reported only two indicators of stress-reduction (decreased level of *demands* and *worries*). Interestingly and in need of further research, levels of α-amylase significantly increased at Site C (built), suggesting that perhaps the setting was efficacious in elevating the sympathetic nervous system, either due to the specific activity or social surrounding.

## 4. Discussion

There is an accumulating body of research from a wide variety of disciplines that suggest that natural environments can have positive effects on human health [4,52]. Defined as an area that is relatively unchanged or undisturbed by human behaviors, natural environments include a broad spectrum of landscapes ranging from wilderness areas, where humans are only short-term visitors, to areas that have been designed, manipulated, or otherwise changed by human interventions. These types of areas typically would include parks, greenspaces, gardens, and waterfront places. A number of pathways exist through which contact with nature may be beneficial to health [53]. A sample of these include improved air quality, increased physical activity, enhanced social contacts, and quality of life.

An important part of this growing corpus of literature concerning human health and natural environments has focused on the construct of “stress”. The presence of stress and its effects on the lives of many people throughout the world is a major health issue in society [54]. Moreover, stress has been linked to a number of physical and emotional issues such as coronary disease, obesity, and depression [55]. Despite this attention to stress and natural environments, a number of questions remain regarding the connection between psychological stress and natural environments, including the type of setting important in stress reduction and more specifically, the question of whether it matters how much nature there is for reducing levels of stress. The results of this study suggest a natural setting can more effectively moderate a visitors’ physiological and psychological stress levels when compared to an urban outdoor setting or indoor exercise facility. After visiting Site A (highest level of nature), visitors’ changes in biophysical markers (i.e., cortisol level) and three dimensions of psychological measures (i.e., levels of demands, worries, and joys) indicated significant decreases in stress levels.

Site A (nature) came closest in providing a location that most represented a wilderness or wildland area. This may be an important consideration due to the powerful emotional and spiritual experiences that are often invoked through a wilderness experience, many of which can have positive health outcomes [53]. This is in line with the argument made by Sato and Conner [56], that the quality of the nature experience can be more important than simply the quantity of the number of experiences. Moreover, this finding is similar to that by Akpinar, Barbosa-Leiker and Brooks [57] who found that the size of a natural environment (i.e., forest in urban areas), was associated with less mental health complaints. A possible explanation of the results of this current study can be explained by Kaplan and Kaplan’s [11] attention restoration theory, where the natural environment is more likely to have factors useful in restoration of attention and reduction of attention fatigue such as fascination, extent, being away, and compatibility. The results of this study support those of earlier findings that have established a positive connection between natural environments and health-related wellness [52,58,59]. It should be noted, however, that McMahan and Estes [60] found no differences in the moderating effects of “wild” nature and “managed” nature on emotional well-being. They posited that managed natural environments such as greenways, green spaces, and arboretums, effectively mimic those characteristics of wild nature that people find appealing, aesthetically pleasing, and restorative. Thus, managed sites may serve as effective substitutes for wild nature. In a similar fashion, Gidlow et al. [21] found that physical exercise had salutogenic effects in both natural and urban environments but that natural environments conferred additional cognitive benefits that could have important connections to reducing variables such as stress. Tyrväinen et al. [23] also found that urban woodlots could be effective in reducing stress levels, even if these visits were short-term. This study adds to the growing corpus of literature that suggests a beneficial effect on reducing levels of stress and that the greater the level of nature the more pronounced the potential benefit is.

## 5. Limitations

There are several limitations present in this study. First, long-term levels of stress, as measured by biomarkers, were not measured [21,33]. While initial (pre) levels of cortisol and α-amylase between the three sites were non-significant, visitors may have come to the respective sites with differing levels of long-term stress, and thus started at different places in their response to stress

Secondly, the interpretation of visitors’ salivary α-amylase levels was limited due to the low effect size and lack of control over different types of activities. According to Nater and Rohleder, [48] and Rohleder, Wolf, Maldonado, and Kirschbaum [42], salivary amylase is more sensitive in reaction to psychological stress or adrenergic activities, and does not seem to be strongly related to other stress biomarkers, such as cortisol. Therefore, further investigation of the relationship between salivary cortisol and salivary amylase changes as well, as their reactions to different types of activities (e.g., aerobic exercise, strength exercise), are warranted.

Third, the participants were not randomly assigned, as may have arrived at the respective sites with different sets of motivations for visitation, sex differences, with males and females often differing in stress responses [36], or how individuals personally interact with various environments. Moreover, although activities engaged in by visitors to Sites A and B were primarily hiking, the same cannot be said of Site C (indoor fitness center). Thus, although the researchers attempted to query participants engaged in running in order to attempt to equalize the types of activities done at each site, the indoor fitness center offered a broader range of specific fitness activities than either Sites A and B. There may be other factors such as types of physical activity, noise, ambiance, or interaction with other people that invoked changes in stress levels. Although beyond the scope of this paper, these confounding variables may have influenced the differences noted between the changes in the levels of cortisol and α-amylase. There is some evidence that suggests that cortisol and α-amylase may be connected to different aspects of the autonomic nervous system [42].

Fourth, although in this study, as per the design of the study, the demographic characteristics such as visitation time and age of the visitor was similar, future studies should use a larger sample with more demographic variance in each cell to examine the possible effects of variables such as sex, age, and frequency of visitation.

Finally, the lack of a qualitative approach in this study precluded the development of a deeper and richer understanding of how the characteristics of these three different sites impacted and were linked to health considerations. Future research efforts should consider including a qualitative approach to provide a better understanding of these underlying factors and the affective meanings visitors attach to various environments.

## 6. Conclusions

Despite these limitations, this study presents findings that lend support for considering visitation to natural environments as potentially useful adjuncts in individuals’ strategy in reducing their risk or as mentioned earlier in this work, their Environmental Reduction Strategy as described by Ryan et al. [61]. Using this framework, the results of this study suggest that the location with the highest level of nature had the greatest effect on reducing levels of stress as measured by biometric and psychometric data. Thus, while individuals may select different locations or activities for reducing stress, for many, natural environments may be useful in their attempts to reduce their levels of stress.

## Figures and Tables

**Figure 1 behavsci-08-00049-f001:**
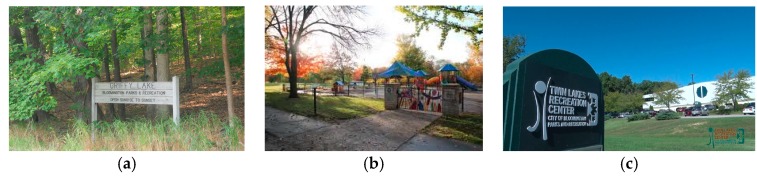
(**a**) Site A (natural setting), (**b**) Site B (semi-natural) (source: visitbloomington.com), (**c**) Site C (urban built).

**Table 1 behavsci-08-00049-t001:** Summary of sample demographics (*n* = 105).

		Site A	Site B	Site C
Gender	Male	18	19	26
	Female	17	16	9
Average Age	Measured by Years	*M =* 25.9	*M* = 37.2	*M* = 28.8
		*SD* = 8.9	*SD* = 15.8	*SD* = 12.8
Average Visit Time	Measured by Minutes	*M* = 54.4	*M* = 68.3	*M* = 66.7
		*SD* = 24.2	*SD* = 27.1	*SD* = 31.1
Frequency of Visit	1–3 times per week	30	17	31
	More than 3 times per week	5	18	4

Data are means ± standard deviations (SDs).

**Table 2 behavsci-08-00049-t002:** Pre- and post-visit comparison of levels of cortisol across the three sites *(n* = 98).

Site	*n*	Baseline	Difference (Post-Pre)	Within-Group Comparison	
				*t*	*p*
Site A	32	3.25 ± 0.27	−0.17 ± 0.30	3.26	<0.01
Site B	31	3.17 ± 0.49	0.04 ± 0.60	0.36	0.72
Site C	35	3.25 ± 0.17	−0.10 ± 0.38	1.50	0.14
ANOVA	*F*	0.67	1.86	-	-
	*p*	0.51	0.16	-	-
Effect Size	*n*^2^ (95% CI)	0.01 (0, 0.07)	0.04 (0, 0.12)	-	-
	ω2 (95% CI)	0.00 (0, 0.05)	0.02 (0, 0.11)	-	-

Data are means ± standard deviations (SDs) in natural Log Scale.

**Table 3 behavsci-08-00049-t003:** Pre- and post-visit comparison of levels of amylase across the three sites (*n* = 104).

Site	*n*	Baseline	Difference (Post-Pre)	Within-Group Comparison	
				*t*	*p*
Site A	34	4.06 ± 0.42	0.02 ± 0.34	0.40	0.69
Site B	35	4.22 ± 0.45	−0.04 ± 0.26	0.93	0.36
Site C	35	4.03 ± 0.33	0.15 ± 0.31	2.79	<0.01
ANOVA	*F*	2.33	3.36	-	-
	*p*	0.10	0.04	-	-
Effect Size	*n*^2^ (95% CI)	0.04 (0, 0.13)	0.04 (0, 0.13)	-	-
	ω2 (95% CI)	0.03 (0, 0.11)	0.03 (0, 0.11)	-	-

Data are means ± standard deviations (SDs) in natural Log Scale.

**Table 4 behavsci-08-00049-t004:** Pre- and post-visit comparison of levels of demands across the three sites (*n* = 101).

Site	*n*	Baseline	Difference (Post-Pre)	Within-Group Comparison	
				*t*	*p*
Site A	33	12.82 ± 2.65	−1.67±3.03	3.16	<0.01
Site B	34	12.91 ± 2.80	−1.32±1.97	3.93	<0.01
Site C	34	11.88 ± 2.31	−1.50±1.96	4.47	<0.01
ANOVA	*F*	1.63	0.18	-	-
	*p*	0.20	0.84	-	-
Effect Size	*n*^2^ (95% CI)	0.03 (0, 0.11)	0.00 (0, 0.04)	-	-
	ω2 (95% CI)	0.01 (0, 0.09)	0.00 (0, 0.02)	-	-

Data are means ± standard deviations (SDs).

**Table 5 behavsci-08-00049-t005:** Pre- and post-visit comparison of levels of worries across the three sites *(n* = 101).

Site	*n*	Baseline	Difference (Post-Pre)	Within-Group Comparison	
				*t*	*p*
Site A	33	10.73 ± 2.84	−2.27 ± 1.59	8.23	<0.01
Site B	34	11.09 ± 2.81	−1.44 ± 1.85	2.93	<0.01
Site C	34	9.59 ± 2.65	−1.18 ± 2.34	4.55	<0.01
ANOVA	*F*	2.71	2.86	-	-
	*p*	0.07	0.06	-	-
Effect Size	*n*^2^ (95% CI)	0.05 (0, 0.15)	0.06 (0, 0.15)	-	-
	ω2 (95% CI)	0.03 (0, 0.13)	0.04 (0, 0.13)	-	-

Data are means ± standard deviations (SDs).

**Table 6 behavsci-08-00049-t006:** Pre- and post-visit comparison of levels of tension across the three sites (*n* = 101).

Site	*n*	Baseline	Difference (Post-Pre)	Within-Group Comparison	
				*t*	*p*
Site A	33	11.36 ± 1.80	−0.30 ± 1.98	0.88	0.39
Site B	34	12.11 ± 2.54	−1.15 ± 2.56	2.61	0.01
Site C	34	12.03 ± 2.08	−0.94 ± 2.07	2.65	0.01
ANOVA	*F*	1.20	1.31	-	-
	*p*	0.30	0.27	-	-
Effect Size	*n*^2^ (95% CI)	0.02 (0, 0.10)	0.03 (0, 0.10)	-	-
	ω2 (95% CI)	0.0 (0, 0.08)	0.01 (0, 0.08)	-	-

Data are means ± standard deviations (SDs).

**Table 7 behavsci-08-00049-t007:** Pre- and post-visit comparison of levels of joy across the three sites *(n* = 101).

Site	*n*	Baseline	Difference (Post-Pre)	Within-Group Comparison	
				*t*	*p*
Site A	33	13.88 ± 2.78	2.30 ± 2.39	5.53	<0.01
Site B	34	14.50 ± 2.62	1.15 ± 2.22	3.02	<0.01
Site C	34	14.79 ± 2.84	0.41 ± 1.71	1.41	0.17
ANOVA	*F*	0.97	6.74	-	-
	*p*	0.38	<0.01	-	-
Effect Size	*n*^2^ (95% CI)	0.02 (0, 0.09)	0.12 (0.02, 0.24)	-	-
	ω2 (95% CI)	0.00 (0, 0.07)	0.10 (0, 0.22)	-	-

Data are means ± standard deviations (SDs).

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
