# Peer review of "Levels of Nature and Stress Response"

_behavsci, 2018, doi:10.3390/bs8050049_

Round 1

Reviewer 1 Report

I enjoyed reading the revised manuscript. A minor suggest goes to the photo of Site A: to keep a consistency with the photos for Site B and Site C, choosing one taken from a normal human eye-level would make more sense.  

Author Response

Responses to Reviewer 1 Comments

Thank you for your suggestion. The photo for Site A has been changed to include an eye-level viewpoint, consistent with Sites B and C.

Reviewer 2 Report

The authors made an accurate process of revision of the original form and my comments/suggestions have been taken in full consideration. I appreciate the extensive revision of the Introduction. Now the goals of the study and its novelty are more clearly defined. Methodological issues have been addressed, as well, and the remaining methodological limits are reported in the Limitations section. Also ethical issues (e.g. lack of informed consent and approval by the Ethic committee) have been resolved.

Still, I do not understand what the following sentence means: “Although less susceptible to issues related to self-report measures …”. In my opinion the “issues” should be described and  a citation should be added here, or the sentence might be cancelled.

Finally, the sentence “Likewise, Hartig and his colleagues [17] compared used a split group of young adults to compared stress recovery…” (lines 86-87), needs a language revision.

Pending these two very minor revisions, my suggestion is to accept the paper in its current form.  My appreciation to the authors for their effort.

Author Response

Responses to Reviewer 2 Comments

To avoid confusion, the sentence related to the issues connected to self-report has been eliminated. (around lines 158-161)

The sentence beginning with “Likewise, Hartig …", has been reworded to provide greater clarity. Thank you for spotting that oversight. (Lines 86-89)